# Bioactive Compounds in Extracts from the Agro-Industrial Waste of Mango

**DOI:** 10.3390/molecules28010458

**Published:** 2023-01-03

**Authors:** Maribel García-Mahecha, Herlinda Soto-Valdez, Elizabeth Carvajal-Millan, Tomás Jesús Madera-Santana, María Guadalupe Lomelí-Ramírez, Citlali Colín-Chávez

**Affiliations:** 1Coordinación de Tecnología de Alimentos de Origen Vegetal, Centro de Investigación en Alimentación y Desarrollo, A.C. (CIAD), Hermosillo 83304, Sonora, Mexico; 2Coordinación de Tecnología de Alimentos de Origen Animal, Centro de Investigación para Alimentación y Desarrollo, A.C. (CIAD), Hermosillo 83304, Sonora, Mexico; 3Departamento de Madera, Celulosa y Papel del Centro Universitario de Ciencias Exactas e Ingenierías, Universidad de Guadalajara, Zapopan 45220, Jalisco, Mexico; 4CONACYT-Centro de Innovación y Desarrollo Agroalimentario de Michoacán, A.C. (CIDAM), Morelia 58341, Michoacán, Mexico

**Keywords:** mango peel, mango seed, mango kernel, bioactive compounds, antioxidant activity

## Abstract

Mango by-products are important sources of bioactive compounds generated by agro-industrial process. During mango processing, 35–60% of the fruit is discarded, in many cases without treatment, generating environmental problems and economic losses. These wastes are constituted by peels and seeds (tegument and kernel). The aim of this review was to describe the extraction, identification, and quantification of bioactive compounds, as well as their potential applications, published in the last ten years. The main bioactive compounds in mango by-products are polyphenols and carotenoids, among others. Polyphenols are known for their high antioxidant and antimicrobial activities. Carotenoids show provitamin A and antioxidant activity. Among the mango by-products, the kernel has been studied more than tegument and peels because of the proportion and composition. The kernel represents 45–85% of the seed. The main bioactive components reported for the kernel are gallic, caffeic, cinnamic, tannic, and chlorogenic acids; methyl and ethyl gallates; mangiferin, rutin, hesperidin, and gallotannins; and penta-O-galloyl-glucoside and rhamnetin-3-[6-2-butenoil-hexoside]. Meanwhile, gallic acid, ferulic acid, and catechin are reported for mango peel. Although most of the reports are at the laboratory level, they include potential applications in the fields of food, active packaging, oil and fat, and pharmaceutics. At the market level, two trends will stimulate the industrial production of bioactive compounds from mango by-products: the increasing demand for industrialized fruit products (that will increase the by-products) and the increase in the consumption of bioactive ingredients.

## 1. Introduction

Mango (*Mangifera indica* L.) is a member of the Anacardiaceae family and is cultivated in more than 90 countries in tropical and subtropical regions, with hundreds of cultivars [1,2,3]. Mango is considered the most popular consumed tropical fruit, containing rich nutritional compounds such as carbohydrates, lipids, and fatty acids, proteins, organic acids, vitamins, minerals, and bioactive compounds. The last are known for their biological properties presented by phenolic compounds (phenolic alcohols, phenolic acids, and flavonoids) and carotenoids (α- and β-carotene), among others [4,5,6,7].

Mango is consumed as fresh fruit, and the pulp is industrially processed as frozen, canned, concentrated, and dried products [1]. Since mango is a flavorful and nutritious fruit, different products, such as juices, jams, jellies, canned slices, mango leather, frozen chunks and slices, pickles, chutney, and mango powder, have been developed [8]. Commercially, mangos are classified into two groups: red varieties (red peel in physiological ripening) and yellow varieties (yellow peel when ripe) [9]. According to the mango variety and its weight, 33–85% constitutes flesh, 7–24% peel and 9–40% seed (*w*/*w*) (Figure 1). Therefore, during mango processing, 35–60% (*w*/*w*) of agro-industrial waste is generated [10,11].

The inappropriate disposal of agro-industrial waste from mango causes environmental problems in the surroundings of the fruit industries [12]. As peels and seeds are important sources of bioactive compounds, their extraction and recovery could help to manage the wastes, while at the same time generating substances with application in the food, pharmaceutical and cosmetic industries, among others. Therefore, the new waste could be seen from a different point of view as a by-product with the potential for valorization. 

In the last decade, many researchers have studied the extraction, identification, and quantification of bioactive compounds from mango agro-industrial waste and their applications. Therefore, the aim of this review was to describe the different investigations (in the last ten years) in which the seed and peel have been used as a source of bioactive compounds, the principal extraction methods, their antioxidant or antimicrobial activities, and the potential to be produced at industrial level.

## 2. Mango Production

According to the Food and Agriculture Organization of the United Nations (FAO), mango production has increased since 2010 (Figure 2A), reaching a value of 41 million tons in 2020. The leading producers are in Asia, Africa, Latin America, and the Caribbean (Figure 2B) [13]. India was the main producer of mango in 2017 with 45.9% of the world’s production [14]. India has the broadest range of mango varieties with high quality and varietal characteristics, and the mango is the national tree [15]. The most important Indian cultivars are Alphonso, Chausa, and Bombay Green [16]. Indonesia was the second mango producer according to FAO (5.9%), with different cultivars, such as Berem, Madu, Gedong, Golek, Bapang, Arumanis, Kepodang, and Kebo [17]. China was the third mango producer (4.3%), with the principal cultivars as Tainong No. 1 and Jinhwang, and these cultivars are characterized by their physicochemical and antioxidant properties [18]. Mexico was the fourth mango producer (4.3%), and the most important cultivars in Mexico are Ataulfo, Kent, and Tommy Atkins [19]. The other mango producers include Pakistan (4.1%), with the Chaunsa as the most popular cultivar, and Brazil (3.6%) where the Ubá variety is a widely produced cultivar [20,21].

## 3. Mango Commercial Varieties

According to the color of the peel, mangos are commercially classified as red and yellow when they reach their physiological ripening. Red mangos are Florida varieties, including Haden, Tommy Atkins, Keitt, Kent, and Palmer. Yellow mango varieties include Totapuri and Alphonso. Red mangos are commonly consumed as fresh fruit and used in the food industry. Furthermore, due to their sensorial characteristics, red varieties dominate the global export market. However, yellow varieties such as Ataulfo or Ubá are preferred because of their nutraceutical properties [9,11,22].

The Haden variety was developed in the United States in 1910. This cultivar is considered a flavored, sweet mango, with many odor notes. For the aroma, Haden has 46 active compounds and thus is considered an aromatic fruit [23]. The Tommy Atkins variety is the most important in the export trade since it preserves its quality along the supply chain, with a shelf life of 8–12 days [24,25]. The Keitt cultivar has primary commercial importance; these mangos are oval and large, with pale to dark green peel, yellow pulp, little fiber, a tangy-sweet flavor with honey notes, and late-ripening [26,27]. The Kent cultivar has a firm pulp and a sweet flavor. Its maturation process is slow and gradual. It is collected near physiological ripening and can be stored for 18 days under marketing conditions (20 °C and 80% relative humidity) [28,29]. The Palmer variety has a firm, yellow flesh and a sweet flavor (21.6 °Brix) pulp with little fiber. Therefore, this variety is consumed as fresh fruit [30].

Yellow varieties such as Totapuri are used for pulp preparation because of their high yield [31]. Alphonso is considered the “King of mangos”: it has a sweet taste, its pulp is non-fibrous, with a golden color, and it is an aromatic fruit. The production of this fruit occurs only in April and May, and its shelf life is limited (7 to 8 days) [32,33]. Ataulfo is a Mexican variety with important organoleptic and sensorial characteristics, and compared with other varieties; it has the highest antioxidant capacity and phenolic compound content [34,35]. Ubá is a variety cultivated in Brazil with excellent sensorial properties, and it is used for developing products such as juice, nectar, and pulp. However, Ubá is consumed only locally because of its fruit size (135.6 g) and peel color [36,37]. 

## 4. Mango Products and By-Products

Mango is consumed as fresh fruit and in different products [38]. The by-products represent 35–60% of the fruit and are constituted by peel and seed (tegument and kernel). The estimated global generation of by-products from mango consumption (fresh and industrialized) after peeling and pitting is between 14.7 and 25.2 million tons per year [39]. Gathering the domestically generated by-products is impractical; however, those industrially generated ones can be perfectly collected. Nearly 0.5% of the world’s mango production is processed [8,39], and the by-products are discarded without treatment, buried with the addition of lime, used for animal feed, used for landfilling or incinerated at an additional cost [40,41]. This agro-industrial waste has high biological oxygen demand and generates environmental contamination. The main problems include the production of unpleasant odors, soil pollution, microbial growth, and the presence of insects and rodents [42]. These problems could be prevented if the industrially generated by-products (71,750–123,000 tons) were collected for further valorization.

## 5. Bioactive Compounds in Mango By-Products

Bioactive compounds are secondary molecules produced in small amounts in vegetables, fruits, grains, and nuts. These molecules show antioxidant, antimicrobial, anticarcinogenic, and anti-inflammatory properties. When consumed, they are considered to be beneficial to our health and have the potential to be extracted and applied as an antioxidant or antimicrobial agent in different fields. Bioactive compounds are classified in essential (vitamins and minerals) and non-essential (polyphenols, flavonoids, carotenoids, phytosterols, glucosinolates, saponins, alkaloids, and essential oils) [43]. One of the most studied topics in the mango agro-industrial by-products is related to the bioactive compounds due to the high content of polyphenols and carotenoids, which have led to many publications focused on their functional properties [9]. Polyphenol molecules are constituted by at least one aromatic ring and at least one hydroxyl group in their structure, and some are present as glycosides. Soluble polyphenols participate in the plant defense system against ultraviolet radiation and attack of pathogens, parasites, and predators. Insoluble polyphenols contribute to the structure of the cell wall to which they are covalently bound. They are classified in phenolic acids such as hydroxybenzoic acids (gallic, salicylic, p-hydroxybenzoic, protocatechuic, vanillic, and syringic acids), hydroxycinnamic acids (ferulic, p-coumaric, chlorogenic, and sinapic acids), and flavonoids (flavonols, flavanones, flavones, isoflavones, and anthocyanidin). Polyphenols are also present in the plant as polymers such as lignin and tannins (gallotannins, ellagitannins, complex tannins, and condensed tannins) [43,44].

Carotenoids are lipid-soluble molecules in the chloroplast and chromoplast of fruits and vegetables. These are yellow, orange, or red pigments that show provitamin A and antioxidant activities and are used as colorants in cosmetic and food products. Carotenoids such as α-carotene, β-carotene, and lycopene are found in carrots, tomatoes, oranges, pumpkins, and mango [43,45]. Figure 3 shows the names and chemical structures of the bioactive compounds reported in mango peels and seeds. Detailed information related to extraction (Table 1), characterization, and some applications of bioactive compounds, found in mango peels and seeds, are included in the following sections.

## 6. Extraction of Bioactive Compounds

Currently, there is no established procedure for the extracting of all bioactive compounds from plant tissues. Thus, many research works have reported the specific procedures used to extract mango wastes. Traditionally, mango peels and seeds are pretreated by drying, homogenization, filtration, or grinding. Next, they are extracted with chemical solvents such as acetone, ethanol, or methanol by techniques such as Soxhlet, percolation, and maceration. More recently, the implementation of physical methods such as pulsed electric fields (PEFs), high-voltage electrical discharges (HVEDs), and ultrasonication allows for the green extraction of polyphenolic compounds using water or ethanol as solvents. Ultrasonication is more efficient than maceration since this technique breaks the cell walls and releases phenolic compounds. Pressurized-liquid extraction can be used to obtain non-polar (fatty acids) and polar compounds (xanthones, phenolic acids, flavonoids, gallate derivatives, and gallotannins). Other processes that assist in the extraction of bioactive compounds are supercritical fluid extraction with carbon dioxide (ScCO_2_) and microwave. The last allows for the higher extraction of phenolic compounds [56,57].

The components of the extracts are separated by chromatographic techniques such as high-performance liquid chromatography (HPLC) equipped with detectors capable of identifying and quantifying them. Mass spectrometry (MS) and diode array (DAD) are mostly used for this purpose. Nowadays, these techniques have evolved into more sensitive instruments with capabilities for identification and quantification at very low concentrations of bioactive compounds in the extracts [57]. However, the traditional technique for quantification of total phenolic compounds (TPCs) by the spectrophotometric Folin–Ciocalteu redox method is widely used. TPC expresses the total antioxidant capacity since high phenolic content has been related to high antioxidant capacity. This technique is based on electron transference and measures the antioxidant capacity in reducing an oxidant, which changes color when reduced. In this case, TPC is reported as mg gallic acid equivalents/g (mg GAE/g). The antioxidant activity of extracts can also be measured by techniques such as Trolox equivalence antioxidant capacity (TEAC), based on the ability of antioxidants to scavenge the radical 2,2′-azonobis(3-ethylbenzothiazoline-6-sulphonate) (ABTS·^+^), the 2,2-diphenyl-1-picrylhydrazyl (DPPH) free radical scavenging method based on the scavenging of the chromogen radical DPPH·, and the ferric ion reducing antioxidant power (FRAP), which measure the ability of antioxidants to reduce a ferric tripyridyltriazine complex, and others [58].

Regarding the extracts containing carotenoids, HPLC with DAD detector is the most frequently used technique for the separation and quantification of this type of bioactive compound. The spectrophotometric technique is also used for the quantification of carotenoids after the removal of interfering substances. Chlorophylls and lipids are eliminated by saponification and the carotene esters are subjected to hydrolysis before quantification. The antioxidant activity of carotenoids is explained because of their electron abundance in the long polyene chain with conjugated double bonds. These are highly reactive and susceptible to electrophilic attack, resulting in stabilized radicals [4,59]. Carotenoids also act as antioxidants by quenching singlet oxygen and excited sensitizers. Thus, the antioxidant activity techniques used for phenolic compounds do not always correlate with the carotenoid content. Therefore, the antioxidant activity of carotenoids is often performed in practical experiments by testing the extracts in vivo or shelf-life experiments [58].

## 7. Mango Peel

### 7.1. Composition of Mango Peel

Mango peel constitutes 7–24% of the whole fruit. It is composed (dry basis) of crude fat (42.24%), carbohydrates (32.31%), crude protein (11.67%), moisture (7.51%), crude fiber (6.32% pectin, cellulose, or hemicelluloses), mineral ash (1.65%), polyphenols and carotenoids, the proportions of which change according to each variety [10,60]. 

### 7.2. Extraction of Mango Peel Bioactive Compounds

In the case of mango peels, two main techniques—cryogrinding (grinding the cooled peel at temperatures <−150 °C) and drying (moisture reduction)—have been studied with the purpose of extraction of bioactive components. Kaur and Srivastav evaluated the cryogrinding effect on the TPC, total flavonoid compounds (TFCs), and half-maximal inhibitory concentration (IC_50_), during mango peel powder obtention in six Indian cultivars. They found that Langra, Neelum, and Chausa cultivars presented high levels of TPC, TFC, and IC_50_ as 60.48 mg GAE/g, 135.04 mg Quercetin equivalents/g, and 490.75 µg/mL, respectively [46]. On the other hand, Oliver-Simancas et al. reported that traditional oven drying at 45 °C for 18 h was an adequate condition to reduce the moisture content while preserving volatile compounds (monoterpenes, sesquiterpenes, C13-norisoprenoids) in peel powder in the Osteen cultivar from Spain. Cryogrinding and drying preserve bioactive compounds in the mango peel powder without additional pre-treatment [47]. Hence, both processes are essential for applications in the industry (mainly in cosmetics).

Sánchez-Camargo et al. extracted carotenoids from the cultivar Sugar from Colombia by ScCO_2_ (conditions of 25 MPa, 60 °C, and 15% *w*/*w* ethanol). They obtained the highest carotenoid content (1.9 mg all-trans-β-carotene equivalent/g dried mango peel), and the extract was used to prevent lipid oxidation in sunflower oil [45]. PEF or HVED extraction combined with water (WE) (25 to 35 °C) allowed for a high recovery of TPC (750 mg/kg HVED, 500 mg/kg PEF, 250 mg/kg WE) and protein content (300 mg/kg HVED, 200 mg/kg PEF, 100 mg/kg WE) from Peruvian mango. PEF or HVED extractions maintained the functional properties, avoiding the use of solvent apart from water, and the best process for extraction was HVED [48]. Another comparison process study was performed between ultrasonication and maceration techniques to extract polyphenols from Pakistani mango peel (Chaunsa var.). Ultrasonication was more efficient in polyphenol extraction, and the principal compounds extracted were gallic acid, ferulic acid, and epicatechin. These compounds were correlated directly with antioxidant activity [49]. The methods are helpful to extract antioxidant compounds in high concentrations from mango peel with solvents such as water and ethanol. The use of ethanol as an extracting solvent would be convenient since it is considered as generally recognized as safe (GRAS) by the USA Food and Drug Administration (FDA) [61]. 

### 7.3. Applications of Mango Peel Extracts

Most of the potential applications of mango peel extracts are related to their antioxidant, antimicrobial, and anti-inflammatory activities, which depend on the variety. Mango peel extracts have been used as components of formulations with potential applications in the film packaging industry for meat, vegetables, and fruits. Kanatt and Chawla developed active packaging films with polyvinyl alcohol (PVA), cyclodextrin and gelatin as polymers, added with ketonic and ethanolic mango peel extracts (Indian var. Langra). Compared with the control, the films exhibited good barrier to UV light, improved mechanical properties, and antioxidant and antibacterial activity. The films were used for the chicken packaging, with an increase in shelf life from 3 to 12 days under refrigeration [62]. Umamahesh et al. studied several varieties of Indian mango peel and concluded that water extracts of the var. Sindhura should be exploited for its components in the development of healthy products. The authors also recommended that eating mango fruit along with its peel would impart more health benefits [63]. The effect of ethanolic extracts from the dried peel of mango (Indian var. Badami) was tested in diabetic rats. The extract contained gallic, protocatechuic, chlorogenic, and ferulic acids. The treated group showed a decrease in diabetic markers and an increase in the activity of different antioxidant enzymes compared with the untreated group. The research concluded that this type of peel extract can be used in functional foods for diabetic persons, whose population is increasing and the demand for food products with antidiabetic properties [64]. The potential application of the components of mango peel as a promoter of wound healing effect was confirmed in a study in which a methanolic extract of mango peel (Mexican var. Ataulfo) was tested in a murine model. The extract contained phenol, resorcinol, and mangiferin, responsible for the antioxidant and anti-inflammatory activities, regulation of oxidative stress, resistance to traction, and activation of the enzymes to produce collagen. Moreover, the extract presented no toxicity, diminished the edema in the animals, and prevented the growth of four bacterial strains associated with wound healing (*Staphylococcus aureus*, *Staphylococcus epidermidis*, *Pseudomonas aeruginosa*, and *Escherichia coli*) [65]. Another application of mango peel extracts (in 70% ethanol) is in animal feeding. Lizárraga-Velázquez et al. formulated a diet for Zebrafish (*Danio rerio*) with different extract concentrations. They found that the fish presented low malondialdehyde concentration in muscle when they were administered with a diet containing 50 and 100 mg/kg of mango peel extract. Moreover, the catalase activity increased with higher concentrations (diet of 150 and 200 mg/kg) of mango peel extract [66]. Regarding applications in food products, chicken sausages were formulated with mango peel extract (obtained with acetone) added in concentrations of 0, 2, 4, and 6%. The sausage’s antioxidant activity (due to polyphenols, anthocyanins, and carotenoids) was proportional to the percentage of the mango peel extract, and the activity was preserved (for 10 days) in a higher level compared with the control. There was no effect of the extract on proximal composition, pH, cooking weight loss, or diameter reduction. However, color was affected at 6% of extract and the recommendation was to add a maximum of 4% of mango peel extract [67].

## 8. Mango Seed and Mango Seed Kernel

### 8.1. Composition of Mango Seed and Mango Seed Kernel

The seed is composed of an endocarp that encloses the kernel, constituting 10–25% of the whole fruit, depending on each variety. The mango seed kernel is inside the seed, representing 45–85% of the seed, and constitutes approximately 20% of the whole fruit [68]. Seed compositional analysis shows values (dry basis) of carbohydrates (78.2%), crude fat (7.8%), moisture (5.1%), crude protein (4.6%), mineral ash (2.2%), and crude fiber (2.1%). Regarding the mango seed kernel, its chemical composition on a dry basis consists of carbohydrates (73.1%), crude fat (9.8%), crude protein (7.2%), mineral ash (2.1%), crude fiber (0.5%), and polyphenols (0.1–8.6 g GAE/100 g). Crude fat is composed of linoleic acid (56.3%) and oleic acid (23.5%), and it is free of trans-fatty acids [69,70,71,72]. Most of kernel fat studies highlight the high oxidative stability due to the presence of a proportion of saturated fatty acids. For instance, the oil extracted from the kernel of mango Dashehari from India (58.08% stearic and 17.99% oleic acids), from twenty Colombian cultivars (37.58% stearic and 46.46% oleic acids), and a Brazilian cultivar (50.69% stearic and 39.04% oleic acids) [73,74,75]. Moreover, the oil extracted from an Iranian mango kernel flour contained lauric (28.7%) and oleic (34.8%) acids [76]. With this fatty acid composition, the oil could be considered stable and tolerant to rancidity and could be used in functional food development. Regarding the bioactive compounds, there are more reports about mango seed kernel extracts than those about mango peel. The kernel oil extract from Indonesian Arumanis mango presented a TPC content of 67.77 mg GAE/g oil and a vitamin E content of 141.22 mg/L. This work showed that mango kernel oil contains natural antioxidants and has application prospects as a bioactive compound [77].

### 8.2. Extraction of Mango Kernel Bioactive Compounds 

Antioxidants are present in mango kernels, and different solvents have been used to extract them. A binary mixture of ethanol and water was used to extract antioxidant compounds from Philippine mangos by solid–liquid extraction. The mixture with 50% (*w*/*w*) ethanol–water had the highest content of TPC (101.68 mg GAE/g of dried mango seed kernel powder). It exhibited the highest antioxidant activity (55.61 and 85.45 Trolox equivalence concentration mmol/L for FRAP and DPPH, respectively). The principal compounds detected in the mango kernel by HPLC-UV/Vis were gallic acid, caffeic acid, rutin, and penta-O-galloyl-β-D-glucose [50]. In another investigation, the TPC and flavonoids of the Egyptian Hindi mango kernel were extracted to give concentrations of 174 mg GAE/g kernel and 33.25 mg catechin equivalent/g kernel, respectively. The principal phenolic compounds detected by HPLC in the mango kernel were hesperidin (30 mg/g kernel), cinnamic acid (12 mg/g kernel), tannic acid (9.86 mg/g kernel), and gallic acid (1.56 mg/g kernel), among others [51]. 

A ternary solvent mixture of methanol–acetone–water (54:23:23) was used to extract antioxidants from mango kernel from Cameroon (11 cultivars). The content of TPC was in the range of 78.22–121.01 mg GAE/g dry weight. The components were separated by high-pressure thin-layer chromatography (HPTLC): chlorogenic acid, mangiferin, gallic acid, and methyl gallate were identified as the principal polyphenolic compounds [44]. Ethanol kernel extracts of two Thai mango cultivars (Keawmorakot and Mahachanock) were studied in two ripening stages. The TPCs were quantified, resulting in 411.77–459.33 mg GAE/g of extract. The ripe Mahachanock extract exhibited the highest antioxidant activity, and HPLC and UV detection systems detected gallic acid the principal phenolic compound in all extracts [52]. Tannins were also extracted with ethanol and methanol from the mango kernel (creole from Perú). The highest content of TPC was in the methanolic extract (963.66 mg GAE/L). Tannins were quantified by a spectrophotometric method, reaching values of 266.96 mg tannic acid/L (after boiling in water for 1 h) and 308.20 mg tannic acid/L (after a 72 h maceration in methanol). The extracted tannins were used in tanning to obtain leather with a good appearance [53].

Pressurized-liquid extraction was used to extract bioactive compounds from mango kernel (Sugar from Colombia) in the non-polar and polar fractions using n-heptane followed by a mixture of solvents (ethanol/ethyl acetate 50:50 *v*/*v*). The non-polar fraction was composed of stearic and oleic acids. The polar compounds found were xanthones, phenolic acids, flavonoids, gallate derivatives, and gallotannins. The gallate derivatives and gallotannins demonstrated in vitro bioactivity when tested for selective antiproliferative activity against human colon cancer cells [54]. Microwave-assisted extraction was used to extract antioxidant compounds from the mango kernel (Ataulfo from Mexico). Response surface methodology showed the optimal values at 1/60 g/mL solid–liquid ratio (ethanol, 90%), 75 °C, and 2 extraction cycles. The antioxidant activity was 1738.2 mg Trolox/g, and the principal antioxidant compounds identified were ethyl gallate, penta-O-galloyl-glucoside, and rhamnetin-3-[6-2-butenoil-hexoside] [55]. The spray-drying technique is one strategy to protect the phenolic compounds in the kernel. Maltodextrin, Arabic gum, and starch were evaluated as encapsulating agents for Philippine mango extracts. The best was maltodextrin, with a yield of 60% at a concentration of 10%. The TPC obtained was 100 mg/g, with an antioxidant activity of 1000 µmol Trolox/g. In addition, there was a low moisture content (3%) and high solubility index (80%), valuable parameters for product development with bioactive compounds that will be reconstituted and dispersed. In general, mango kernel antioxidants are extracted with solvents such as water, ethanol, and methanol, and the concentration changes according to the cultivar studied. The extraction method influences the extracted compounds by yield, stability, and concentration according to the solvent used [78]. 

Due to the composition and proportion, extraction of bioactive compounds from mango seed kernel has been explored more than that from peels, mainly at the laboratory level. The following section also shows several applications reported for kernel extracts.

### 8.3. Applications of Bioactive Extracts from Mango Kernel

Most applications of mango kernel extracts have been reported in the field of active packaging. An active film packaging with antioxidant properties obtained by the casting process was developed with combinations of mango kernel starch, mango kernel fat, and phenolic extract obtained from the Tommy Atkins cultivar from Brazil. The best films were produced with kernel starch and phenolic extract without fat (because of the high saturation of fatty acids) since they exhibited relatively high antioxidant activity, UV-absorbing capacity, and good barrier properties. The films could be used in foods with a high lipid content [79].

Soy protein isolate and fish gelatin were used for developing active packaging containing extract from Malaysian mango kernel, added in concentrations from 1% to 5% as a natural antioxidant. The films were produced by the casting technique. The presence of the kernel extract in the films increased the thickness, tensile strength, antioxidant activity, and reduced the water solubility and elongation at break. The films showed their potential as active packaging, leaving applications in food products susceptible to oxidation for future studies [80]. In addition, the soy protein isolate film added with the extract was evaluated regarding its temperature stability (−18, 4 and 25 °C) for 90 days. At all the studied temperatures and different times, the only changes observed during the test were a darker yellow color and a decreased elasticity. At 25 °C, the concentration of phenolic compounds was higher than at −18 and 4 °C, and the antioxidant activity remained stable for all tested storage times. At 25 °C, there was an increase in the protein–protein interactions, and the antioxidant film activity involved migration of antioxidants from the film to the product. The higher temperature promoted a higher re-arrangement process in the protein matrix, and at lower temperatures, the re-arrangement process decreased [81].

Mango peel flour (Mexican var. Alphonso), added with an ethanolic extract from the same variety of mango seed kernel, was used to elaborate edible films by the cast technique. Control and antioxidant films were tested by TPC (14.7 and 44.2 mg GAE/g, respectively) and DPPH˙ radical scavenging activity (52.9 and 63.5%, respectively). The coating application was conducted by immersing peaches in solutions of the control and antioxidant films in water. The addition of the seed kernel extract resulted in decreased permeability, improved color, antioxidant properties, and hydrophobicity. The coating with the antioxidant solution contributed to a reduction in gas transfer (O_2_, CO_2_, and ethylene) and demonstrated an extension in the shelf life of peaches from 4 to 8 days at 20 °C [82].

The antimicrobial activity of mango kernel extracts has also been studied and applied in antimicrobial active packaging, such as that formulated with kernel extract from the Australian mango (var. Pearl), on printed latex coatings. The extract contained phenolic compounds that can be used against different bacteria of fish and meat. The authors reported that films were produced with a commercial multilayer food packaging (Cryovac^®^ BLUS 50, SealedAir, Hamilton, New Zealand), with a layer of polyethylene in contact with food) coated with the extract powders dissolved in aqueous carboxylated styrene-butadiene. The film with mango kernel extract presented zones of microbial growth inhibition in disk diffusion assays for all bacteria tested (16 common spoilage and pathogenic bacteria such as *Pseudomonas fluorescens*, *Salmonella* Enteriditis, and *Listeria monocytogenes*, among others), and this performance is essential to produce commercial films and the potential applications in perishable foods such as fish and meat [83].

Mango kernel extracts also present antimicrobial activity with the potential for water disinfection. A study evaluated the effect of mango seed kernel extracted with sodium chloride solutions (0.8–100 mg/mL) for water treatment against *Escherichia coli*, *Salmonella typhi*, *Klebsiella* spp., *Pseudomonas* spp., *Enterococcus faecalis*, and *Staphylococcus aureus*. The results suggested that extracts in the agar diffusion method were more effective at 100 mg/mL. In the broth dilution method, the extracts were bacteriostatic at a low concentration (*E. coli* 50 mg/mL, *S. typhi* 50 mg/mL, *Klebsiella* spp. 12.5 mg/mL, *Pseudomonas* spp. 3.1 mg/mL, *E. faecalis* 25 mg/mL, and *S. aureus* 25 mg/mL) and bactericidal at high concentrations (*E. coli* 100 mg/mL, *S. typhi* 100 mg/mL, *Klebsiella* spp. 50 mg/mL, *Pseudomonas* spp. 12.5 mg/mL, *E. faecalis* 50 mg/mL, and *S. aureus* 100 mg/mL). In addition, the antimicrobial activity of extracts was similar to that of chlorine, and they could be used as a green material for the disinfection of water at low cost [84].

Fermentation of mango seed and kernel resulted in increased antioxidant activity. A mango seed (25%) and mango kernel (75%) mixture (Ataulfo from Mexico) was used as a substrate for solid-state fermentation with *Aspergillus niger*, which induces the structural breakdown of plant cell walls, leading to the liberation or synthesis of various antioxidant compounds. The free phenolic fraction of fermented mango kernel seed increased from 984 to 3288 mg GAE/100 g at 20 h of fermentation. The phenolic fraction of fermented mango contained hydrolyzed compounds from gallotannins related to antioxidant activity, and the fermented extract can be used as a natural antioxidant for the pharmaceutical and food industry [85].

Another study implemented a high-speed homogenization technique to create solid lipid nanoparticles. The purpose was to improve the stability of bioactive chemicals extracted with ethanol from a Thai mango kernel, with application in the cosmetic industry. The best formulation was composed of an extract concentration of 60 mg/100 mL, Tween 80 fraction of 1%, and 1:2 extract:lipid ratio. The nanoparticles were stable for 15 days at 4 °C without evident changes in clarity and degradation [86].

The mango residues can also be used as a biocontrol agent. Mango peel and mango seed extracts with antioxidant and anti-yeast properties were tested against clinically pathogenic (*Candida bracarensis*, *Candida glabrata*, *Candida nivariensis*, and *Candida parapsilosis)* and food-spoilage yeasts (*Dekkera anomala*, *Hanseniaspora uvarum*, *Zygosaccharomyces bailii*, *Zygosaccharomyces bisporus*, and *Zygosaccharomyces rouxii*). Three cultivars (Keitt, Sensation, and Gomera-3 from Spain) were used, and nine extracts were obtained. All extracts presented anti-yeast activity against the tested species. However, the minimum inhibitory concentration and the minimum fungicidal concentration were lower for the seed extracts than for the peel extracts, with differences between cultivars. The anti-yeast activity was correlated with the TPC since the mango residue extracts contained proanthocyanidins (0.08–0.80 mg leucoanthocyanidin equivalents/100 g dry weight), gallates and gallotannins at high concentrations [87].

## 9. Perspectives

Fresh fruits are not always available because they are produced in a season of the year or due to the difficulties of a long-distance transportation, as it occurs with tropical fruits. Moreover, in some countries, fruit production has diminished due to climate change phenomena [88]. Fortunately, industrialization makes possible the global availability of tropical fruits. The frozen fruit market is the sector with the highest growth rate broadcast, with 6.7% from 2020 to 2027. Mango is among the frozen tropical fruits that accounted for the largest share (>40%) in 2019 [89]. The dry fruit sector is expected to grow at 5.2% from 2020 to 2025 [90]. With respect to mango juice, the market of fruit juices, mainly those made from orange, apple, mango, mixed fruits, and others, is expected to grow at 2.1% from 2021 to 2026 [91]. Therefore, these trends will contribute to increasing the number of by-products from the mango industry over the next few years. Moreover, the trend to reduce industrial wastes as one of the principles of the circular economy is boosting to scale up the already reported laboratory processes for valorizing the agro-industrial by-products from mango. The techno-economic viability of integrated mango processing waste biorefineries has been published to valorize mango produced in India and South Africa [92,93]. They used process simulation software based on results obtained at the laboratory level for biorefinery models. Three approaches for mango by-products were evaluated in India: only pectin extraction from mango peels, pectin and seed oil extraction, and whole biorefinery with multiple product recoveries, which included the extraction of polyphenols. They found that the three cases could offer opportunities for developing processing systems in a circular economy, generating wealth from mango wastes if the plant was located close to the mango processing plant. The whole biorefinery with multiple product recoveries resulted in the most convenient option [92]. Manhongo et al. evaluated the process feasibility, economic viability, and environmental impact in three scenarios for mango by-product valorization, where the principal activity was the dried mango chips in South Africa. The scenarios were: an integrated biorefinery for the anaerobic digestion of mango peel and wastewater for biogas production, the second integrated the extraction and recovery of pectin from mango peel, and the third involved the sequential recovery of pectin and polyphenols coupled with heat and electricity production. The results showed that the third scenario was the most capital-intensive option and presented the highest environmental impact; however, it was the most favorable in terms of profitability. Both viability studies with Indian and South African mango showed a negative effect when the waste processing plant was idle out of the mango production season. Therefore, the alternative of drying mango waste during the highest harvesting season will increase availability and help to maintain the plant working longer. Another option is to use the facilities to process other fruit wastes out of the mango production season [93]. Regarding patents, three related to the valorization of mango by-products were found. The first is about mango seed oil skin-protecting detergent; the second is about the development of paper products and methods of making paper products with antimicrobial properties with mango seed extract; and the third is about the preparation of pectin and polyphenolic compositions from mango peel [94,95,96]. The preceding informs about the research needs and opportunities for the comprehensive use of mango residues. With respect to products in the market, there are cosmetic products based on mango seed kernel fat such as soaps, body lotion, lip balms, etc. 

On the other hand, the demand for bioactive ingredients is expected to grow at 4.8% from 2022 to 2029. This increase is due to the rising consumption of functional foods because of the growing awareness about health and nutrition. Additionally, the COVID-19 pandemic has enhanced this industry, as more bioactive compounds are researched for managing, treating, and preventing this illness. The advances in extraction technologies, such as pressurized-liquid extraction, are replacing conventional solvent extraction technology, allowing for rapid extraction with low quantities of solvents. Other examples of new technologies implemented at the industrial level are pressurized-liquid extraction, subcritical/supercritical extractions, and microwave- and ultrasound-assisted extractions [97].

Considering the perspectives mentioned above, the broadcast of growth for the sectors related to the generation of fruit wastes, demand for bioactive compounds, and development of advanced technologies suitable for industrial use is positive. At the laboratory level, enough studies have been published, informing the composition of mango by-products and potential applications in fields such as food, cosmetics, oil and fat, active packaging, and pharmaceutic industries. Meanwhile, only some studies explored the use of modern extraction technologies or the viability of bioactive compounds extraction at the industrial level. The extraction of only bioactive compounds from mango by-products may not be economically feasible, and most of the biomass will remain a waste. However, the sequential processes for the valorization of not only the minor but the major components of these by-products (pectin, starch, fat, and fiber) [98] have proved to be technically and economically feasible at the industrial level. The feasibility increases if other fruit/vegetable wastes are processed using the same equipment and facilities to maintain the plant production in all seasons. More studies evaluating the commercial viability of the industrialization of mango by-products are needed to generate objective information for the interested sectors.

## 10. Conclusions

The global production of mango reached 41 million tons in 2020, from which 0.5% is industrialized as juices, jams, jellies, canned slices, mango leather, frozen chunks and slices, pickles, chutney, and mango powder, among others. During mango processing, 35-60% of the fruit is discarded, in many cases without treatment, generating environmental problems and economic losses. The by-products of the industrial processes are peel and seed (kernel and tegument), reaching 123,000 tons annually. In the last ten years, the main bioactive compounds reported in mango by-products are polyphenols (high antioxidant and antimicrobial activities) and carotenoids (provitamin A and antioxidant activity), among others. The kernel has been studied more than peels and tegument. The main bioactive components reported for kernel are gallic, caffeic, cinnamic, tannic, and chlorogenic acids; methyl and ethyl gallates; mangiferin, rutin, hesperidin, and gallotannins; and penta-O-galloyl-glucoside and rhamnetin-3-[6-2-butenoil-hexoside]. Meanwhile, for mango peel, gallic and ferulic acids and catechin are reported. With respect to the tegument, there are no reports of bioactive compounds. Most of the reports are at the laboratory level, and only a few reported the feasibility on the industrial scale to show concrete opportunities to decrease the mango waste. They have potential applications in food, active packaging, cosmetics, oil and fat, and pharmaceutic industries. The processes with the highest potential resulted in being those which integrate the recovery of several components such as pectin and polyphenols coupled with heat and electricity production and whole biorefinery with multiple product recoveries. The need for studies to scale up the laboratory processes to the industrial level is essential to generate objective information for the interested sectors. Lastly, the broadcasting for the industrialization of fruits is expected to grow up from 2.1% to 2.6% in the following years, increasing the production of by-products.

## Figures and Tables

**Figure 1 molecules-28-00458-f001:**
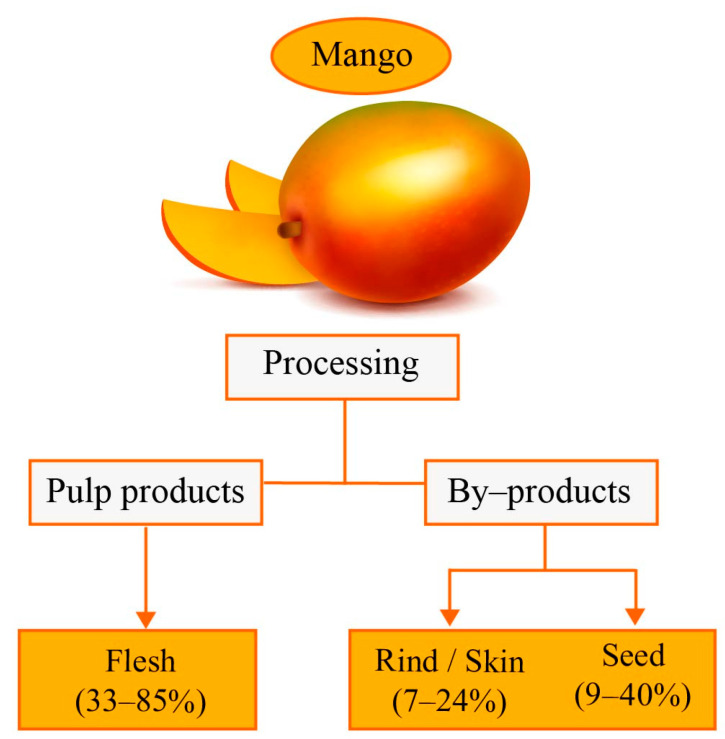
Proportions of flesh and by-products produced during mango processing [10].

**Figure 2 molecules-28-00458-f002:**
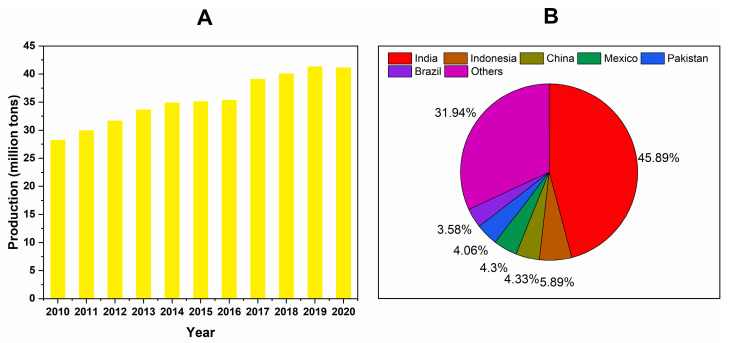
Worldwide mango production from 2010 to 2020 (**A**) and leading producers in 2020 (**B**). Source: Food and Agriculture Organization of the United Nations [13].

**Figure 3 molecules-28-00458-f003:**
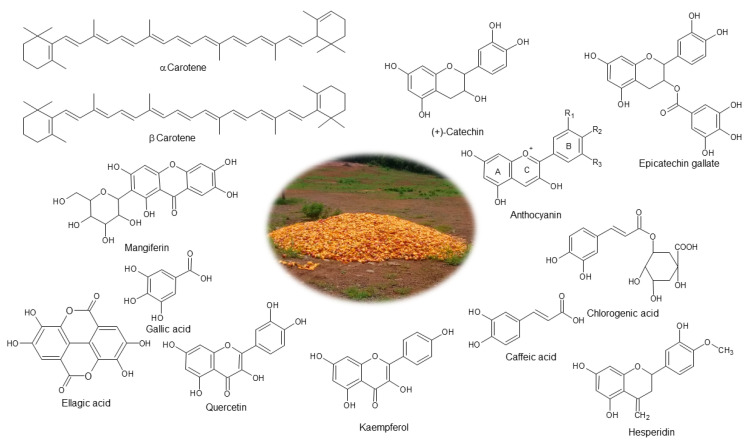
Chemical structure of bioactive compounds found in mango agro-industrial by-products.

**Table 1 molecules-28-00458-t001:** Methods for extraction of bioactive compounds from mango by-products.

By-Product	Method	Process or Extractant	Quantification Technique (Bioactive Compounds)	Reference
Peel	Cryogrinding	Moisture reduction	TFC and TPC	[46]
	Drying	Moisture reduction	Volatile compounds	[47]
	PEFs and HVDEs	Water	TPC and proteins	[48]
	Pressurized Extraction	ScCO_2_	Carotenoids	[45]
	Ultrasonication and Maceration	Ethanol and methanol	TPC (gallic and ferulic acid, epicatechin)	[49]
Kernel	Solid—Liquid Extraction	Ethanol and water	TPC (rutin and penta-*O*-galloyl-β-D-glucose	[50]
	Solid—Liquid Extraction	Methanol	TFC and TPC (hesperidin, cinnamic, tannic acid)	[51]
	Solid—Liquid Extraction	Methanol-acetone- water	TPC (chlorogenic acid, mangiferin, methyl gallate)	[44]
	Solid—Liquid Extraction	Ethanol	TPC (gallic acid)	[52]
	Solid—Liquid Extraction	Methanol and ethanol	Tannins	[53]
	Pressurized—Liquid Extraction	Ethanol	TPC (xanthones, phenolic acids, flavonoids, gallate derivatives and gallotannins)	[54]
	Microwave—Assisted Extraction	Ethanol	TPC (ethyl gallate, penta-*O*-galloyl-glucoside and rhamnetin-3-[6-2-butenoil-hexoside])	[55]

TFCs: total flavonoid compounds; TPCs: total phenolic compounds; PEFs: pulsed electric fields; HVEDs: high-voltage electrical discharges; ScCO_2_: super critical fluid extraction with carbon dioxide.

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
