# Peer review of "Bioactive Compounds in Extracts from the Agro-Industrial Waste of Mango"

_molecules, 2023, doi:10.3390/molecules28010458_

Round 1

Reviewer 1 Report

1. The study period covered (10 years, 15 years, 20 years) in the review should be clearly stated in the Abstract and Conclusion.

2. The review contains three figures, however, there is no any tabular data. It will be very useful to summarized the data and present in Tables. Separate tables for components from different parts of mango.

Author Response

Reviewer 1

We thank the Reviewer for his/her suggestions and critics. See the responses:

  1. Lines 24 and 528: the period of the study covered 10 years. Added to the abstract and conclusions sections.
  2. Table 1 was added to section 6.

Reviewer 2 Report

the manuscript is quite interesting for the readers, especially for the separation process. however, several comments are addressed to improve the quality of the manuscript. 

1. please revised the affiliation number

2. please add the objective of the review paper in the introduction part

3.  proofread and format editing should be done 

4.  in lines 157 to 165, please combine in one paragraph

5.  please add the nomenclature part for this manuscript

6. in line 204 to 215, please combine in one paragraph

7.  add more applications of mango peel extracts

Author Response

Reviewer 2

We thank the Reviewer for his/her suggestions and critics.

  1. The affiliation numbers were corrected.
  2. Lines 22 and 63-66: the aim of the review was added to the abstract and introduction sections.
  3. Proofreading was done, and the format was edited.
  4. Lines 154-161: the paragraphs were combined as suggested.
  5. Lines 555-576: an abbreviation list was added before the References section.
  6. Lines 199-210: the paragraphs were combined as suggested.
  7. Lines 264-295: more applications of mango peel were added.

Author Response

Reviewer 3:

We thank the reviewer for his/her suggestions and critics. We sent the manuscript for publication because we received an invitation to contribute to a special issue named: Bioactive compounds and antioxidant activity of extracts from different natural plants. There was no request of presenting scaling or feasibility studies.  However, we understand the worries of the reviewer and to respond we added a new section of perspectives (9), in which we included references related to the feasibility of the recovery of mango by-products, the market trends related to the demand of industrialized fruit (dried, frozen and juice), including mangos and their by-products. Also, we presented the increasing demand of bioactive ingredients for different sectors and the new technologies that are replacing the traditional solvent extraction processes. We hope the reviewer agrees with the new information. Please see section 9 (lines 462-520), abstract (lines 33-36) and conclusions (lines 536-544).

We also changed the title of the manuscript as follow: 

Line 2. The title of the manuscript was changed to “Bioactive compounds in extracts from the agro-industrial waste of mango”.  Thank you for the suggestion.

Round 2

Reviewer 3 Report

The authors have modified their manuscript to a certain extent, but they have not critically assessed the prospects for commercially isolating natural products from mango waste. To my mind this is a missed opportunity to provide an objective assessment of the current state of the field and to provide a constructive critique of the current gaps and suggest how the field should develop in the future. With a bit more time and effort the authors could have produced a much more mature and insightful manuscript; as it is, it reads as though it was essentially left to the most junior author to write it.  The newly added paragraphs are vague, lack specific proposals and make no effort to assess the commercial viability in terms of future demand for the products, the industrial cost of isolating the natural products and bringing them to market and, of course, the environmental cost of not processing the waste.

Author Response

We thank the reviewer for her/his critics and suggestions that motivated us to improve the manuscript. We have rewritten the last paragraph in section 9 (lines 512-527). 

In response to the comment: “...no effort to assess the commercial viability in terms of future demand for the products, the industrial cost of isolating the natural products and bringing them to market and, of course, the environmental cost of not processing the waste”, we included the following sentences: “The extraction of only bioactive compounds from mango by-products may not be economically feasible, and most of the biomass will remain a waste. However, the sequential processes for the valorization of not only the minor but the major components of these by-products (pectin, starch, fat, and fiber) have proved to result in technical and economically feasible to be produced at the industrial level. The feasibility increases if other fruit/vegetable wastes are processed using the same equipment and facilities to maintain the plant production in all seasons” (lines 519-524).

In the case of the request “… constructive critique and suggestions how the field should develop in the future”, we wrote: “more studies evaluating the commercial viability of the industrialization of mango by-products are needed to generate objective information for the interested sectors” (lines 525-527).

In the Conclusion section we added: “The needs for studies to scale up the laboratory processes to the industrial level is essential to generate objective information for the interested sectors.

In response to the “Moderate English changes required”, the manuscript was subjected to a grammar review with a considerably improvement.